# SARS-CoV-2 Spike Protein and Neutralizing Anti-Spike Protein Antibodies Modulate Blood Platelet Function

**DOI:** 10.3390/ijms24065312

**Published:** 2023-03-10

**Authors:** Boguslawa Luzak, Marcin Rozalski, Tomasz Przygodzki, Magdalena Boncler, Dagmara Wojkowska, Marcin Kosmalski, Cezary Watala

**Affiliations:** 1Department of Hemostasis and Hemostatic Disorders, Medical University of Lodz, 92-215 Lodz, Poland; 2Department of Clinical Pharmacology, 1st Chair of Internal Diseases, Medical University of Lodz, 90-153 Lodz, Poland

**Keywords:** SARS-CoV-2 spike protein, COVID-19, anti-SARS-CoV-2 IgG antibodies, blood platelets, platelet reactivity, platelet adhesion, platelet activation

## Abstract

Several studies report elevated blood platelet activation and altered platelet count in COVID-19 patients, but the role of the SARS-CoV-2 spike protein in this process remains intriguing. Additionally, there is no data that anti-SARS-CoV-2 neutralizing antibodies (nAb) may attenuate spike protein activity toward blood platelets. Our results indicate that under in vitro conditions, the spike protein increased the collagen-stimulated aggregation of isolated platelets and induced the binding of vWF to platelets in ristocetin-treated blood. The spike protein also significantly reduced collagen- or ADP-induced aggregation or decreased GPIIbIIIa (fibrinogen receptor) activation in whole blood, depending on the presence of the anti-spike protein nAb. Our findings suggest that studies on platelet activation/reactivity in COVID-19 patients or in donors vaccinated with anti-SARS-CoV-2 and/or previously-infected COVID-19 should be supported by measurements of spike protein and IgG anti-spike protein antibody concentrations in blood.

## 1. Introduction

Several studies have shown elevated blood platelet activation and changes in platelet count in COVID-19 patients [1,2,3,4]. Platelets from subjects with severe forms of SARS-CoV-2 infection demonstrate increased surface exposure of P selectin (CD62) [3] in their resting state or upon activation with TRAP or 2MeSADP [5] and active form of GPIIbIIIa complex [4]. Moreover, thromboxane A2 release [5] and platelet-leukocyte aggregate formation were observed [3,5]. Platelet aggregation was significantly increased in non-severe and severe COVID-19 patients in response to low-dose agonists (2MeSADP, thrombin, and collagen) and showed increased spread on both fibrinogen- and collagen-coated surfaces [5,6]. Furthermore, COVID-19 plasma, added to the blood of healthy subjects, induced similar platelet activation to that observed in vivo in COVID-19 patients [7]. In addition, IgG antibodies from patients with severe COVID-19 are able to stimulate FcƴRIIA, leading to the induction of procoagulant platelets with increased thrombus formation ability [8].

SARS-CoV-2 can be found in circulation more abundantly than previously thought, and it was observed that plasmatic viremia correlates with disease severity and mortality [9]. Several studies have confirmed the presence in blood plasma or serum of nucleocapsid antigen [10,11,12,13] or spike protein as a whole protein molecule or its S1 subunit [9,14]. The presence of circulating spike protein in the blood of COVID-19 patients with post-acute sequelae for up to 12 months post-diagnosis was reported, and this may suggest that SARS-CoV-2 viral reservoirs may persist in the body [15]. Additionally, circulating SARS-CoV-2 proteins were found in the plasma of participants vaccinated with the mRNA-1273 vaccine from Moderna [16], and circulating exosomes expressing spike protein on day 14 after vaccination with the mRNA-based SARS-CoV-2 vaccine (BNT162b2, Pfizer-BioNTech) followed by a significant increase in concentration at day 14 after vaccination with the second dose were reported [17]. The presence of a wide range of fragmented SARS-CoV-2 RNA in platelets from patients with COVID-19, direct uptake of the virus by platelets, and the digestion of SARS-CoV-2 in platelets making it non-infectious were demonstrated [18]. SARS-CoV-2 internalization triggers platelet death programs that cause platelet content to leak and thus reduce their functionality [18].

The mechanism of blood platelet interaction with the SARS-CoV-2 virus remains controversial [19,20]. The key factor for virus attachment to target cells is the spike protein, the main SARS-CoV-2 membrane glycoprotein. The spike protein forms homotrimers that protrude from the viral surface; it comprises two functional subunits which facilitate viral attachment to the surface of the host cell (S1 subunit) and fusion of the viral and cellular membranes (S2 subunit) [21]. The spike protein in the S1 subunit contains an RGD motif (arginine–glycine–aspartate) near the distal tip of its receptor-binding domain with structural features reminiscent of known integrin-binding proteins [22]. Conflicting data suggests that the main cell entry receptor for SARS-CoV-2, angiotensin-converting enzyme 2 (ACE2), may be absent in blood platelets [2,5,6,19,23]. Alternative receptors that could bind SARS-CoV-2 and promote its entry into platelets are CD147 (basigin and EMMPRIN) [24] and the integrins GPIIbIIIa or GPIb [22,25,26]. When antibodies against SARS-CoV-2 are present or cross-reacting antibodies against more prevalent coronaviruses that cause minor cold symptoms, viruses can also activate platelets through indirect interactions with FcRIIA [27].

Understanding the immunological response to SARS-CoV-2 and the role of anti-SARS-CoV-2 antibodies should extend our knowledge of the mechanisms of COVID-19. Contact with the SARS-CoV-2 virus or vaccination results in the production of neutralizing antibodies (nAb) that contain an epitope for RBD. The vaccines authorized in Europe display a good safety profile in clinical trials; however, rare incidences of vaccine-induced immune thrombotic thrombocytopenia or thrombosis with thrombocytopenia syndrome have been reported [28]. In the case of a virus-vectored vaccine, it is evident that antiPF4/polyanion antibody is directly involved in the pathogenesis of thrombosis [29]. It is also possible that the spike protein-induced inflammation may lead to thrombosis [28]. The vaccination-mediated adverse effects can be attributed to the unique characteristics of the spike protein itself (antigen), either due to molecular mimicry with human proteins or as an ACE2 ligand [30]. Klug et al. analyzed the expression of constitutive transmembrane receptors, adhesion proteins, and platelet activation markers in 12 healthy donors before the first BNT162b2 administration and then at another five-time points within four weeks; the findings reveal that BNT162b2 administration does not alter platelet protein expression and platelet reactivity [31].

Bearing in mind the potential prothrombotic activity of spike protein and its presence in blood during COVID-19 infection or after anti-SARS-CoV-2 vaccination, the aim of the present study is to determine whether the presence of anti-spike protein antibodies offers protection to blood platelets against the deleterious effects of the spike protein. We hypothesize that a high level of anti-spike protein antibodies in the blood of healthy donors cancels the activating effect of spike protein on platelet reactivity.

Our findings indicate reduced platelet aggregation in whole blood and decreased activation of GPIIbIIIa (receptor for fibrinogen) after incubation with spike protein and that this effect was more profound in donors with a high level of anti-spike protein nAb. However, spike protein was found to induce vWF binding to platelets in ristocetin-treated blood.

## 2. Results

### 2.1. Characteristics of Study Participants

Healthy donors were recruited between January 2022 and June 2022. Before blood collection, every participant was asked about the vaccination for SARS-CoV-2. Of all participants, 17 (32%) donors reported receiving three doses of the COVID-19 vaccination, 22 (42%) donors had received two doses, five (9%) had received one dose, and nine (17%) had not received any SARS-CoV-2 vaccination. The vaccinated participants had received the Comirnaty vaccine (Pfizer-BioNTech), AstraZeneca, Moderna, or Janssen vaccine. Almost all study participants had anti-SARS-CoV-2 TrimericS IgG (nAb) concentration over 33.8 BAU/mL (positive), and only three donors from the total group were negative for anti-SARS-Cov-2 IgG (below 33.8 BAU/mL). Additionally, over 50% of donors had more than 2080 BAU/mL. In the total group, the morphology parameters (blood cell count, hemoglobin, hematocrit, and MPV) and C-reactive protein (CRP) concentration were in the normal diagnostic reference range. The characteristics of the participant group are shown in Table 1. In order to investigate the role of anti-spike protein neutralizing antibodies (nAb) on platelet function, the total group was divided into two subgroups based on the median of anti-SARS-CoV-2 TrimericS IgG concentration: a High Ab-group (above 1115 BAU/mL) and Low Ab-group (below 1115 BAU/mL). The two subgroups demonstrated similar age, BMI, MPV, blood cell counts, and inflammatory marker levels (Table 1). No significant differences in platelet activation/reactivity were found between the groups when aggregation in whole blood (Figure 1B,C), CD62 exposure (Figure 2B,C), and fibrinogen binding to platelets (Figure 3B,C) were analyzed. However, significantly increased fibrinogen receptor (GPIIbIIIa) activation (PAC1 binding) was observed for resting platelets (*p* < 0.05), TRAP- (*p* < 0.01), or collagen-induced (*p* = 0.07) platelets in the High Ab-group (Figure 4B,C). Furthermore, significantly increased PAC1 binding for ADP-induced platelets in blood stimulated with agonists at high concentrations was found in the High Ab-group (Table 2; *p* < 0.05; Appendix A).

### 2.2. Platelet Reactivity in the Presence of Spike Protein Depending on the Level of Anti-Spike Protein Neutralising Antibodies

Platelet activation and reactivity were monitored in the presence of spike protein using several indicators: platelet aggregation, the surface exposure of P selectin (CD62), the activated form of GPIIbIIIa (PAC1 binding), endogenous vWF or exogenous fibrinogen binding, and clot retraction time. The effect of spike protein on platelet reactivity was analyzed depending on the concentration of anti-spike protein antibodies (High and Low Ab-groups).

#### 2.2.1. Aggregation and Activation of Platelets

The incubation of whole blood with spike protein resulted in a significant reduction in collagen- or ADP-induced aggregation (*p* < 0.001) in the High Ab-group. However, spike protein was not found to have any effect in the Low Ab-group or on TRAP-induced aggregation. In the total group, the effect of spike protein was similar to that in the High Ab-group (Figure 1). 

Flow cytometry analysis revealed a significant reduction in GPIIbIIIa activation (PAC1 binding) in ADP-stimulated platelets for the samples incubated with spike protein in both the High and Low Ab-group and also in the total group (Figure 4). In both subgroups, no significant differences were found between spike and control samples for a fraction of PAC1-positive platelet activated with TRAP or collagen. Likewise, the spike protein did not significantly modulate the activation of unstimulated platelets or platelet reactivity, monitored as P selectin (CD62) exposure (Figure 2) or the binding of exogenous fibrinogen to platelets (Figure 3). In addition, the analysis of the fluorescence intensity (median values) for PAC1-positive platelets confirmed the reduction effect of spike protein in ADP-stimulated platelets with statistical significance in the Low Ab-group and in the total group (Appendix A). On the contrary, spike protein significantly enhanced CD62 exposure in TRAP-induced platelets in the High Ab-group but not in Low Ab-group. In the total group, the spike protein effect was the same as in the High Ab-group (Appendix A). The level of the fibrinogen bound to platelets (estimated as median fluorescence) was similar irrespective of the concentration of nAb or type of agonist (Appendix A).

The results obtained using aggregation and flow cytometry methods (measured in whole blood in both techniques) revealed that the effect of spike protein on platelet reactivity can be modulated by anti-spike protein nAb present in whole blood. Additionally, this effect varies for different types of agonists. In TRAP-activated platelets, spike protein had no effect, and similar results were obtained for the Low and High Ab-groups. In ADP-stimulated platelets, decreased platelet reactivity (whole blood aggregation and activation of fibrinogen receptor GPIIbIIIa) was noted in the presence of a spike, with the highest inhibition for the High Ab-group. For collagen-induced platelets, a significant reduction in whole blood aggregation was observed in all participants and in the High Ab-group; spike protein had no effect on CD62 exposure or on the exogenous fibrinogen binding, irrespective of anti-spike protein nAb concentration or agonist type. Moreover, spike protein had no significant effect on surface CD62 exposure, activation of GPIIbIIIa complex, and fibrinogen binding for non-activated platelets. 

Whole blood aggregometry showed significantly reduced collagen- or ADP-stimulated platelet reactivity in the presence of the spike protein. Therefore, the next step monitored the aggregation of platelets in platelet-rich plasma (PRP) (LTA method) from donors with high anti-spike nAb concentrations. Using this method could allow us to observe the interaction between platelets and spike protein in plasma (if present) with the exclusion of erythrocytes and leukocytes. No significant differences were observed between control and spike protein samples with regard to ADP- and collagen-stimulated aggregation (69.2 ± 25.0 vs. 66.2 ± 27.3% for ADP; 78.6 ± 7.1 vs. 71.8 ±15.9% for collagen; in control vs. spike respectively, n = 5). The representative curves are presented in Appendix A.

#### 2.2.2. Clot Retraction

In the Low Ab-group, but not in the High Ab-group, a shorter time to halving the apparent thrombus area in the presence of spike protein was demonstrated (Figure 5). The same observation as in Low Ab-group was for the total participant group (Figure 5A). The weight of clots after 120 min of retraction time did not significantly differ between spike and control samples (8.9 ± 2.8 mg for spike vs. 9.4 ± 4.4 mg for control, n = 37).

### 2.3. Aggregation of Isolated Platelets

In further experiments, we investigated the interaction of the spike protein with blood platelets in isolated conditions where the plasma and blood cells were absent. To evaluate these direct effects, washed (isolated) platelet aggregation induced by either collagen (1 µg/mL) or ADP (2 µM) or thrombin (0.1 U/mL) was monitored after incubation with 2 µg/mL spike protein (15 min, at 37 °C). Significantly increased platelet aggregation was observed under the influence of spike protein when collagen was used as an agonist (Figure 6), but not for ADP or thrombin: aggregation for control and spike treatment was 12.3 ± 12.2% and 12.9 ± 10.9% (n = 18) for ADP and 81.5 ± 10.2% and 85.5 ± 6.3% (n = 8) for thrombin. Additionally, spike protein did not affect aggregation when used alone as a stimulator (2.3 ± 2.2% for control vs. 4.0 ± 3.4% for spike, n = 4). Spike protein had no effect when CD62 exposure or PAC1 binding was measured (Appendix A).

### 2.4. Monitoring of the GPIb Role in Spike Protein Interaction with Blood Platelets

To investigate a potential mechanism underlying the spike protein–blood platelet interactions, we analyzed the role of GPIb and the binding of vWF to platelets in the presence of spike protein.

#### 2.4.1. Adhesion under Flow Conditions

The platelet adhesion to vWF in the presence of spike protein was unaltered (*p* = 0.0713), with a tendency to a higher level compared to the control (in five out of six donors) (Figure 7). Furthermore, adhesion to fibrinogen was also overall unaltered in spike protein-treated blood samples (*p* = 0.0904) compared to control (Figure 7). In the group of donors investigated in the adhesion experiments, the level of nAb was various (ranging from 15 to >2080 BAU/mL).

#### 2.4.2. The Role of GPIb and vWF Binding in the Interaction of Spike Protein with Blood Platelets

In the next step, the role of platelet glycoprotein Ib (GPIb) and the plasmatic vWF binding to this receptor in the interaction with the spike was monitored. It was then shown that the spike protein did not change the level of platelet surface-bound vWF (in the vWF-positive fraction of platelets) in either resting (non-agonist) or TRAP-induced platelet samples (Figure 7). Nevertheless, ristocetin treatment resulted in a significantly increased number of vWF-positive platelets in blood in the presence of spike protein (*p* < 0.05). In addition, the level of the plasmatic vWF binding to platelets increased in the presence of spike protein in ristocetin-treated samples (Appendix A). Preincubation of ristocetin-treated blood with anti-GPIb antibodies (AK2) completely blocked the binding of vWF to the platelet surface irrespective of the presence of spike protein (*p* < 0.05; Figure 8). The experiment was performed on the study participants with various levels of anti-spike protein (ranging from 15 to >2080 BAU/mL).

As it was shown earlier, the spike protein significantly reduced collagen-induced platelet aggregation in whole blood. Preincubation with AK2 antibodies did not significantly influence collagen-stimulated platelet aggregation (Table 3). A significant reduction in aggregation was observed in the presence of spike protein in both control samples (whole blood with saline) and in AK2-treated samples. These experiments were conducted in blood from donors with extremely high levels of anti-spike protein antibodies (above 2080 BAU/mL).

### 2.5. Thrombin Generation Assay in Plasma

The parameters of thrombin generation, such as lag phase and thrombin concentration, were monitored in platelet-poor plasma (PPP) and in platelet-rich plasma (PRP). In both cases, plasma was incubated with spike protein for 15 min at 37 °C before the induction of thrombin generation by adding the phospholipid micelles to PPP or tissue factor (TF) to PRP. It was observed that in vitro administration of spike protein to PPP samples did not affect the TG parameters (Appendix A). Additionally, thrombin concentration or lag time was not influenced by the fibrinogen concentration in plasma or by the level of anti-spike nAb, irrespective of the presence of spike protein (correlations were weak and statistically insignificant). In PRP, where blood platelets were present during the reaction, the presence of spike protein only slightly modulated thrombin concentration, with no effect on lag time (Appendix A). It seems that the addition of tissue factor to PRP may influence thrombin level in the spike-treated samples: TF at 1 pM increased thrombin generation to a greater degree than TF at 2.5 pM, or 5 pM compared to samples without TF; however, the differences are not significant.

## 3. Discussion

The mechanisms underlying increased platelet activation in COVID-19-infected patients are intriguing. Endothelial disruption and cytokine storms result in a prothrombic state, including stimulation of platelets and other blood cells; in addition, it has been found that the SARS-CoV-2 or its proteins are internalized by platelets, which may suggest the direct platelet-virus interaction. A few scientific reports describe the direct modulation of platelets by SARS-CoV-2 or spike protein [23,24,25,32]. However, Puhm et al. report that platelets are not activated by SARS-CoV-2 or by the purified spike protein but rather by TF derived from extracellular vesicles released from monocytes. Similarly, it appears more likely that platelet hyperactivation is caused by inflammatory stress associated with SARS-CoV-2 infection by low concentrations of SARS-CoV-2 in the circulation via direct virus-platelet interaction [32]. Zhang et al. clearly show that spike protein significantly increases platelet activation/reactivity in an ACE2 receptor-dependent manner [23]. They note that the spike protein of SARS-CoV-2 isolated from a COVID-19 patient and propagated in Vero E6 cells directly enhanced platelet activation (PAC1 binding, CD62P exposure, α granule secretion, and dense granule release) and aggregation, platelet spreading, and clot retraction in vitro. The effects were more evident for isolated platelet previously incubated with spike protein (2 µg/mL for 5 min) or SARS-CoV-2 (1 × 10^5^ PFU, for 30–60 min) and then stimulated with collagen (0.6 μg/mL), thrombin (0.025 U/mL), and ADP (5 μM) [23]. Our findings indicate only a slight increase in collagen-induced aggregation in isolated platelets (up to 10%) in the presence of spike protein (2 µg/mL, 15 min 37 °C), with no influence from ADP- or thrombin-induced aggregation. Moreover, CD62 exposure or active fibrinogen receptor GPIIbIIIa exposure on isolated platelets stimulated with collagen or ADP remained unchanged after incubation with spike protein. In addition, spike protein did not appear to have any stimulatory effect on platelets in whole blood with regard to P selectin exposure or PAC1 binding or in platelet-rich plasma following ADP- and collagen-stimulated aggregation. However, spike protein significantly elevated plasmatic vWF binding to platelet in ristocetin-treated blood, which suggests that the spike protein may influence the interactions between platelets and vWF. These findings are in agreement with those of Ruberto et al. [33], who report markedly increased levels of vWF antigen and the active form of vWF binding to platelets (vWF:RCo) in COVID-19 patients. These results were associated with higher agglutination rates induced by ristocetin, thereby indirectly indicating an increased capability of vWF to bind to platelets [33]. They also note that in the case of severe COVID-19, platelets bind more easily to plasmatic vWF and that no correlation was found between ristocetin sensitivity and vWF concentration in patient blood [34]. Importantly, the relationship between platelets and vWF is very important in primary hemostasis. In platelets, vWF may not only bind to GPIbα (CD42b), a part of the GPIb-IX-V complex but also to GPIIbIIIa, the receptor for fibrinogen, thus resulting in platelet adhesion, aggregation and clot formation [35]. COVID-19 patients demonstrate significantly elevated levels of antigen and activity of vWF and mildly to moderately-reduced ADAMTS-13 activity [35]. In addition, autopsies of COVID-19 patients who died of acute respiratory distress syndrome revealed numerous intrapulmonary arteriole thrombi, including fibrin, CD61-positive platelets and megakaryocytes, with positive immunostaining of vWF [36]. 

The potential mechanism of the increased platelet-vWF interactions observed in COVID-19 patients or in vitro is difficult to explain. It is believed that SARS-CoV-2 spike protein may interact with GPIbα and facilitate vWF-GPIbα binding. Li et al. report that SARS-CoV-2 can activate platelets directly and identified GPIbα as the binding receptor for spike protein, although the binding affinity was only moderate [25]. They also indicate that S-RBD (spike receptor-binding domain) inhibited ristocetin-induced recombinant vWF binding on isolated platelets, which may suggest the competitive antagonism of GPIbα by the spike protein. Our findings indicate that AK2 antibodies, which block vWF binding to GPIbα, reduced the plasmatic vWF binding to platelets in ristocetin-treated blood irrespective of the presence of spike protein. These results showed that spike protein interacts with platelets in the site differently from the vWF binding site in platelet membrane GPIbα.

We observed that spike protein treatment was found to accelerate clot retraction, suggesting the presence of prothrombotic properties: the time taken to reduce the clot to half its area was significantly shorter in the participants with the low level of anti-SARS-CoV-2 neutralizing antibodies (below 1115 BAU/mL). However, thrombin generation in plasma or in platelet-rich plasma did not differ between the high and low nAb groups and was not influenced by the presence of spike protein. These results suggest that a high level of anti-SARS-CoV-2 Ab could reduce the effect of spike protein in clot stabilization. Clot retraction is a process driven by outside-in signaling of platelet integrin αIIbβ3 (GPIIbIIIa) that results in the contraction of the fibrin mesh in a thrombus [37], which increases clot density and decreases clot size. Outside-in signaling of the αIIbβ3 receptor is induced by fibrin or fibrinogen binding to the receptor and plays a key role in clot retraction. Thus, a clot becomes smaller, and excess fluid is extruded. Clot retraction is a physiologically important mechanism allowing the close contact of platelets in primary hemostasis, the reduction in wound size, and reperfusion in case of thrombosis [38]. However, the greater platelet number and fibrin crosslinking have the overall effect of increasing fibrin density and reducing clot permeability and pore size, which affects fibrinolysis. Increased thrombin generation and decreased fibrinolysis were reported in COVID-19 patients in vivo [39]. In another study of COVID-19 patients, pronounced significant inhibition of all stages of blood clot contraction was observed. The impaired clot contraction was associated directly with disease severity and poor outcomes. In patients with severe disease, blood clot contraction was suppressed significantly compared to patients with moderate disease manifestations [40]. Globbelaar et al. showed that spike protein might interfere with blood flow. It may cause structural changes in β and γ fibrin(ogen), complement 3, and prothrombin, which results in these proteins being substantially resistant to trypsinization [41].

Hypercoagulability is one of the main characteristics of COVID-19, but a few papers report reduced platelet reactivity in SARS-CoV-2 infected patients [5,33,42]. Ruperto et al. found that platelet aggregation in plasma (PRP) in response to both ADP and collagen was lower in COVID-19 patients than in healthy volunteers; however, no significant differences in platelet aggregation were observed in response to arachidonic acid [33]. Manne et al. describe increased platelet reactivity in COVID-19 patients, manifested as elevated platelet activation and increased aggregation in response to 2MesADP, thrombin, and collagen, as well as increased adhesion and spread of platelets on fibrinogen and collagen during infection with SARS-CoV-2. Interestingly, they found that in whole blood from COVID-19 patients, platelet activation by 2MeSADP, TRAP, and collagen-related peptide resulted in a decreased PAC1 binding, irrespective of disease status, compared to healthy donors. They suggest that the decreased GPIIbIIIa activation was not due to changes in αIIb expression, as far as the αIIb expression was similar among healthy donors and all COVID-19 patients [5]. Furthermore, in the study of Li et al. in collagen-induced PRP samples incubated with spike protein, it was demonstrated lower luminescence derived from ATP/ADP release compared to those treated with collagen alone [25]. In turn, Jacobs et al. demonstrated higher TRAP-, ADP-, and AA-stimulated platelet reactivity measured by whole blood aggregometry in COVID-19 compared with patients with other acute respiratory diseases [1]. The results from platelet reactivity in COVID-19 are hence very varied, depending on COVID-19 patient characteristics, disease status, control group characteristics, and experimental protocols. Heinz et al. [43] found platelet aggregability to be unchanged in COVID-19 patients in response to TRAP or arachidonic acid but to be reduced by ADP treatment compared to healthy subjects; however, they conclude that it is not possible to generalize on platelet function in COVID-19 [43]. This lower platelet reactivity may be due to platelets possibly being extensively activated in vivo during COVID-19; this may result in refractoriness to new agonists added during ex vivo platelet function tests, a phenomenon called *exhausted platelets* [42]. Another hypothesis is that platelet reactivity may be modulated by serum/plasma components from COVID-19 patients and by IgG anti-SARS-CoV-2 antibodies [7,8,44]. Therefore, it has been suggested that the potential for thrombus formation may be increased via crosslinking FcƴRIIA by the Fc domain of IgG antibodies in a calcium-dependent manner [8]; this may also be true for immobilized immune complexes containing recombinant anti-spike IgG with low fucosylation and high galactosylation, enhancing thrombus formation on vWF [44]. 

Pelzl et al. found that incubation with sera or IgG increased the generation of procoagulant platelets from COVID-19 patients and that this is mediated by IgG antibodies through the PI3K/AKT signaling pathway in an FcγRIIA-dependent manner [45]. Our present findings suggest that platelet reactivity may be modulated not only by anti-SARS-CoV-2 nAb but particularly by anti-spike protein nAb/spike protein complexes. Neither significant platelet activation nor reactivity was found to differ between the low Ab (nAb < 1115 BAU/mL) and high Ab (nAb > 1115 BAU/mL) groups; however, activation of GPIIbIIIa (PAC1 binding) was significantly increased in the high Ab group. This observation may suggest that the presence of spike protein-neutralizing antibodies in the blood of healthy participants following vaccination or prior infection may promote the procoagulant state of platelets in the early phase (activation of GPIIbIIIa). However, this effect is not significant for further platelet activation (fibrinogen binding, aggregation, and degranulation). Interestingly, the presence of spike protein resulted in a significant reduction in ADP- or collagen-induced aggregation in whole blood and in the level of PAC1 binding in ADP-stimulated platelets. The results were related to the concentration of anti-spike protein nAb in blood, and the highest inhibitory effect was found for donors with an nAb level higher than 1115 BAU/mL. These findings are in accordance with those of Manne et al., Bertolin et al., and Ruberto et al., who identified reduced platelet reactivity in COVID-19 patients [5,33,42]. Our present findings cannot be attributed to *exhausted platelets*, as the study only enrolled healthy volunteers without hemostatic or inflammatory problems. Additionally, spike protein was added to the blood before measurements, and it is likely that the decreased platelet aggregation with ADP and collagen and decreased PAC1 binding (expression of activated GPIIbIIIa form) on ADP-stimulated platelets is triggered by anti-SARS-CoV-2 Ab/spike protein complex. 

These results are the first to indicate that reinfection with SARS-CoV-2 or anti-SARS-CoV-2 vaccine booster, when the anti-spike protein nAb/spike protein complex can be generated, may modulate platelet reactivity. Interestingly, the interaction between platelets and the anti-spike protein nAb/spike protein complex disturbs the inside-out signaling for GPIIbIIIa activation, reducing aggregation and suppressing the expression of the active GPIIbIIIa form on platelets. Further studies are needed to identify the platelet receptor involved in this process. It cannot be excluded that P2Y12 or GPVI may interact with the complex because ADP- or collagen-stimulated platelet reactivity was modulated by spike protein in vitro or in COVID-19 patients with active SARS-CoV-2 infection. While the FcRIIa receptor could also be engaged, this is less likely because the interaction of the antibody Fc region with cellular Fc receptors results in platelet activation but not inhibition.

## 4. Materials and Methods

### 4.1. Chemicals

Recombinant spike protein (rSARS-CoV-2 B.1.1.7 (GCN4-IZ) His-tag, Cat. No. 10796-CV) was from R&D Systems (Minneapolis, MN, USA). Collagen, thrombin, ristocetin, glass cuvettes, and stirring bars were purchased from Chrono-Log (Havertown, PA, USA). Antibodies anti-human CD61/PerCP (clone RUU PL 7F12), CD61/PE (clone VI-PL2), CD62P/PE (clone AC1.2), PAC1/FITC (IgM κ-immunoglobulin), mouse IgG1/PE isotype control, mouse IgM/FITC isotype control, Cellfix, and BD Vacutainer Systems with buffered sodium citrate or acid-citrate-dextrose/glucose (ACD solution A) were purchased from Becton-Dickinson (Franklin Lakes, NJ, USA). RPE-conjugated anti-human CD41 antibodies (clone 5B12) were from Agilent Technologies (Santa Clara, CA, USA). Anti-human von Willebrand Factor antibodies IgG1/FITC (clone IIIE2.34) and mouse IgG1/FITC isotype control were from Novus Biological (Abingdon, UK). Human vWF native protein and fibrinogen from Human Plasma Oregon Green 488 Conjugate were purchased from Invitrogen (Carlsbad, CA, USA). Anti-human CD42b (GPIb) antibodies (clone AK2) were from Thermo Scientific (Regensburg, Germany). Phosphate buffered saline pH 7.4 (PBS) was obtained from Corning (New York, NY, USA). TRAP (SFLLRNPNDKYEPF, Thrombin Receptor Activator Peptide), dimethyl sulfoxide (DMSO), adenosine diphosphate (ADP), HEPES, and bovine serum albumin (BSA) were obtained from Sigma (St. Louis, MO, USA). Native fibrinogen from human plasma was from Calbiochem (Darmstadt, Germany). TGA Technothrombin reagent C Low and thrombin calibrator were from TC Technoclone (Vienna, Austria). Fluorogenic substrate 2GGA-AMC was from Bachem (Bubendorf, Switzerland). All other chemicals, unless otherwise stated, were purchased from Avantor Performance Materials Poland S.A. (Gliwice, Poland).

### 4.2. Study Participants and Blood Collection

For experiments conducted using whole blood, platelet-rich plasma (PRP), or platelet-poor plasma (PPP), blood was collected from healthy donors (n = 53; 47% men and 53% women; mean age 32.7 ± 9.5 years) into a vacuum tube containing 0.105 mol/L buffered sodium citrate (final citrate: blood ratio of 1:9 *v*/*v*); for experiments requiring isolated platelets, the blood was collected into a vacuum tube containing acid citrate dextrose (ACD) (final ACD: blood ratio of 1:9 *v*/*v*). For clot retraction analysis, the blood was collected into a vacuum tube containing 4% *w*/*v* sodium citrate (final citrate: blood ratio of 1:9 *v*/*v*). 

All participants stated that they had not taken medications known to influence platelet function for at least two weeks prior to the study. The following were the exclusion criteria: infection, recognized cancer disease, anemia, coronary artery disease, chronic inflammatory disease, liver disease, and pregnancy. A blood sample was taken for every participant to determine the blood cell count, hemoglobin, hematocrit, mean platelet volume (MPV), and C-reactive protein (CRP) concentration. Laboratory parameters were measured using standard diagnostic procedures. Platelet, neutrophile, and lymphocyte counts, and MPV were used to calculate the inflammatory rates: PLR (platelet-to-lymphocyte ratio), NLR (neutrophil-to-lymphocyte ratio), and MPVLR (MPV-to-lymphocyte ratio).

Fresh serum samples were used to measure SARS-CoV-2 IgG concentration with the highly-effective LIAISON^®^ SARS-CoV-2 TrimericS IgG assay; this enables reliable detection of IgG antibodies against Trimeric S spike protein (Diasorin S.P.A, Saluggia, Italy). All samples were processed by trained diagnostic laboratory staff according to the manufacturer’s procedures, using the specified controls and calibrators. The LIAISON^®^ SARS-CoV-2 TrimericS IgG assay is a quantitative test that meets the WHO International Reference Standard. The test measures values between 4.81 and 2080 BAU/mL, with a cut-off level of 33.8 BAU/mL. The manufacturer reported a clinical specificity of 99.5% (95% CI: 99.0–99.7%) and clinical sensitivity of 98.7% for samples ≥15 days post-RT-PCR confirmation.

Plasma fibrinogen level was measured quantitatively using Dia-FIB reagent according to the manufacturer’s protocol (Diagon, Budapest, Hungary).

Experiments were approved by the Ethics of Research in Human Experimentation Committee at the Medical University of Lodz (approval No. RNN/08/22/KE), and all participants gave their informed written consent to participate in the study.

### 4.3. Determination of Platelet Aggregation in Whole Blood

Whole blood impedance aggregometry was carried out using a five-channel multiplate analyzer (Roche Diagnostics GmbH, Mannheim, Germany) according to the manufacturer’s guidelines. After incubation of whole blood with a spike protein (2 µg/mL) or saline (control) for 15 min at 37 °C, platelet aggregation was monitored for six minutes following the addition of collagen (0.5 µg/mL), ADP (2 µmol/L), or TRAP (2 µmol/L). Maximal aggregation (Amax) was chosen as the evaluation criterion.

### 4.4. Monitoring of Platelet Surface Activation Markers with Flow Cytometry: P-Selectin (CD62P) Exposure, GPIIb-IIIa Activation (PAC1-Positive Platelet Fraction), Exogenous Fibrinogen Binding, and Plasmatic vWF Binding to Platelet Surface

Whole blood was preincubated with a spike protein (2 µg/mL) for 15 min at 37 °C. Whole blood preincubated with diluent (saline) was used as a control sample. The platelets were activated with 2 µM ADP, 2 µM TRAP, or 2 µg/mL collagen for 5 min at RT. In another series of blood samples without spike protein, platelets were activated with agonists at high concentrations (10 µM ADP, 8 µM TRAP, or 2 µg/mL collagen for 5 min at RT). 

After activation in both experimental protocols, the samples were diluted 10-fold with PBS, labeled with anti-CD61/PerCP, anti-CD62P/PE, and PAC1/FITC antibodies (15 min, RT), and fixed with CellFix for one hour at RT. Directly before measurements, the samples were diluted 1:1 with PBS, and the assay was performed, gathering 10,000 CD61/PerCP-positive events using a FACS Canto II flow cytometer (BD Bioscience, Franklin Lakes, NJ, USA). In the gated population of CD61/PerCP-positive events, the percent fractions of the platelets positive with regard to given surface membrane markers (above isotype cut-off) were measured. Moreover, the abundances (the numbers of copies) of given surface membrane markers were presented as a median of fluorescence intensity values (MFI values).

The binding of exogenous fibrinogen was monitored in whole blood preincubated with a spike protein (2 µg/mL) or saline (control) for 15 min at 37 °C. Oregon Green-labeled fibrinogen was added to the blood samples (30 µg/mL), which were subsequently activated with 2 µM ADP, 2 µM TRAP, or 2 µg/mL collagen for 5 min at RT. The samples were then diluted 10-fold with PBS, labeled with anti-CD61/PE antibodies (15 min, RT), and fixed with CellFix for one hour at RT. Directly before measurement, the samples were diluted 1:1 with PBS, and the assay was performed, gathering 10,000 CD61/PE-positive events using a FACS Canto II flow cytometer (BD Bioscience, Franklin Lakes, NJ, USA). The percent fractions of fibrinogen-positive platelets and the relevant median MFI values were measured in the gated populations of CD61/PE-positive events.

Similar to other protocols, the expression of vWF on the platelet surface was measured in whole blood preincubated with a spike protein (2 µg/mL) or saline (control) for 15 min at 37 °C. After incubation, the blood samples were treated with ristocetin (2 mg/mL) or with TRAP (10 µM) for 5 min at RT. In the ristocetin-induced samples, AK2 antibodies were used to block the interaction between plasma vWF and its receptor GPIb (preincubation at 37 °C for 10 min, followed by incubation with the spike protein). Next, blood samples were diluted 10-fold with PBS, labeled with anti-CD61/PE, and anti-vWF/FITC or isotype control IgG1/FITC (15 min, RT), and fixed with CellFix for one hour at RT. Directly before measurements, the samples were diluted 1:1 with PBS, and the assay was performed, gathering 10,000 CD61/PE-positive events using a FACS Canto II flow cytometer (BD Bioscience, Franklin Lakes, NJ, USA). The percent fraction of vWF-positive platelets (above isotype cut-off) and the relevant median MFI values were measured in the gated population of CD61/PE-positive events. 

### 4.5. Assessment of Clot Retraction

Clot retraction was analyzed according to Tucker et al. [37] with a minor modification. The blood was centrifuged at 190× *g* for 12 min at 37 °C, and platelet-rich plasma (PRP) was collected. A portion of PRP was centrifuged at 2000× *g* for 10 min to obtain platelet-poor plasma (PPP). For the clot tests, the platelet count was adjusted to 3 × 10^8^ platelets/mL with PPP. Finally, human thrombin (25 µL) was added to a final concentration of 1 U/mL to the warmed (37 °C) mixture of buffer (134 mM NaCl, 0.34 mM Na_2_HPO_4_, 2.9 mM KCL, 12 mM NaHCO_3_, 20 mM HEPES, 5 mM glucose, 1 mM MgCl_2_, and pH 7.3; 262 µL), PRP (200 µL), red blood cells (3 µL of autologous erythrocyte pellet); this mixture was supplemented with 2 µg/mL spike protein in the study group or saline in controls. The assay was conducted within an incubator at 37 °C. Photographs were taken at 0, 15 min, and then every 30 min until 120 min. The images were analyzed to provide numerical data, such as apparent thrombus area, using FIJI image analysis software [46]. Finally, data were presented as a time to half-reduction in the thrombus area. After 120 min, the clots were weighed.

### 4.6. Monitoring of Platelet Adhesion under Flow Conditions

The experiments were performed using the VenaFlux platform (Cellix, Dublin, Ireland). Depending on the experimental setup, the channels of a Vena8 Fluo+ biochip were coated with human fibrinogen (100 µg/mL) or with human von Willebrand factor (100 µg/mL) overnight at 4 °C and blocked with bovine serum albumin (1 mg/mL) for one hour at 4 °C. The biochip was mounted on a thermally-controlled stage of an inverted AxioVert microscope (Zeiss, Jena, Germany) at a constant temperature of 37 °C. Prior to measurements, the channels were washed with saline. Blood was collected as described above. To test the effect of spike protein, the samples were preincubated at 37 °C for 15 min, either with the protein at a final concentration of 2 µg/mL or an equivalent volume of saline. Prior to perfusion through the channels, the samples were recalcified by the addition of CaCl_2_ to a final concentration of 2 mM; they were then passed for 2 min through the microchannels at a shear rate of 445 s^−1^ (20 dynes/cm^2^) on the fibrinogen channels or at 890 s^−1^ (40 dynes/cm^2^) on the vWF channels. The channels were then washed with Cellfix solution (100 s^−1^, 2 min) to remove nonadherent platelets and incubated for one hour at RT to fix the adherent cells prior to labeling. Firmly adherent blood platelets were stained for 30 min with anti-CD41/RPE antibodies. Visualization and imaging were performed with an epifluorescence microscope (Xyloxemine, Zeiss, Jena, Germany) in ten different fields along the channel. The areas covered by the platelets were quantified by means of FIJI software [46], preceded by thresholding using Ilastik software [47].

### 4.7. Monitoring of Platelet Aggregation in Platelet Rich Plasma

Light transmission aggregometry (LTA) was performed in PRP. PRP was prepared by 12-min of 200× *g* centrifugation of blood at 37 °C. PPP was obtained by centrifuging PRP samples at 2000× *g* for 15 min at 37 °C. The PRP samples, with a final platelet count of 2.2 × 10^8^ cells per ml, were incubated with spike protein (2 µg/mL) or saline (control) for 15 min at 37 °C. After incubation, platelet aggregation was monitored for 10 min following the addition of collagen (1 µg/mL) and ADP (2 µmol/L) on a dual-channel aggregometer (Chrono-Log, Havertown, PA, USA). Maximal aggregation (Amax) was used as the outcome.

### 4.8. Determination of Isolated Platelet Function in the Presence of Spike Protein

To monitor platelet aggregation, the blood was centrifuged with prostaglandin E1 (1 µmol/L) at 200× *g* for 12 min at 37 °C to obtain PRP. The collected PRP was centrifuged at 700× *g* for 15 min, and the obtained sediment suspended in Tyrode buffer (134 mM NaCl, 12 mM NaHCO_3_, 2.9 mM KCl, 0.34 mM Na_2_HPO_4_, 1 mM MgCl_2_, 10 mM HEPES, 5 mM glucose, pH 7.4) with ACD solution A (9: 1) and prostaglandin E1 (1 µmol/L), and centrifuged at 700× *g* for 15 min. After centrifugation, the blood platelets were suspended in Tyrode buffer and adjusted to a count of 2 × 10^8^ platelets/mL with a Sysmex XS-800i^TM^ automated hematology analyzer (Sysmex, Kobe, Japan). The isolated platelets were incubated with spike protein (2 µg/mL) or with saline for 15 min at 37 °C. After incubation, LTA was performed using a dual-channel aggregometer (Chrono-Log, Havertown, PA, USA) with collagen (1 µg/mL), thrombin (0.1 U/mL), and ADP (2 µM) preceded by the addition of fibrinogen at a final concentration of 2 mg/mL. Results were reported as maximum aggregation (%) 10 min after adding the agonist.

Flow cytometric analysis was performed for the gel-filtered platelets. In this protocol, blood samples were collected into vacutainer tubes containing ACD solution A (BD Vacutainer System, Franklin Lakes, NJ, USA) as an anticoagulant at a volume ratio of 9:1; the mixtures were then centrifuged to obtain PRP. After the platelet suspension was collected from a CNBr-activated Sepharose 4B, the platelets were adjusted to 2 × 10^8^/mL with a Tyrode buffer. The isolated platelets were incubated with spike protein (2 µg/mL) or with saline (as a control) for 15 min at 37 °C. Next, the platelets were activated with 10 or 20 µg/mL collagen or 5 or 10 µM ADP for 5 min at RT. All samples were then labeled with anti-CD62P/PE and PAC1/FITC antibodies (15 min, RT). Flow cytometry was performed using a BD LSR II flow cytometer (Becton Dickinson, Franklin Lakes, USA) and analyzed with BD FACSDIVA 6.0 software. The platelets were gated using anti-CD61 antibodies. The percentage of CD62- or PAC1-positive platelets (above isotype cut-off) was measured. At least 10,000 cells were analyzed per sample.

### 4.9. Thrombin Generation

Thrombin generation (TG) was measured in PRP or in PPP prepared from blood collected in sodium citrate (9:1) and centrifuged for PRP (12 min, 200× *g* at 37 °C) or for PPP (15 min, 2000× *g* at RT). In the case of PRP, thrombin generation was tested in freshly-prepared platelet suspensions of a density of 2 × 10^8^/mL. In both PPP and PRP, the plasma samples were first incubated with spike protein (2 µg/mL) or with saline (control) for 15 min at 37 °C. The samples were transferred to a 96-well fluorescence plate. Next, thrombin generation was initiated using different approaches for PPP and PRP. In PPP, TG was measured using a mixture of phospholipid micelles containing rhTF (Technothrombin TGA reagent C Low). In PRP, the tissue factor was added to plasma at a final concentration of 1–2.5–5 pM. 

Following this, for both PPP and PRP, 1 mM fluorogenic substrate 2GGA-AMC in a HEPES buffer (20 mM HEPES, 150 mM NaCl, pH 7.35) with 0.5% BSA and 15 mM CaCl_2_ was added, and the fluorescence measurements were begun in a Victor™ X4 plate reader (Perkin-Elmer, Turku, Finland). The thrombin generation parameters were evaluated automatically using the TECHNOTHROMBIN^®^ TGA evaluation software (www.technoclone.com). The thrombin calibration curve was measured individually, according to the instructions. Two main parameters were used in the final analysis: the lag time (the clotting time, i.e., the moment at which TG begins) and the peak height, i.e., the maximum thrombin concentration.

### 4.10. Statistical Analysis

Data were presented as mean ± SD or median and interquartile range (IQR), depending on the normality of data distribution. The Shapiro–Wilk test and Levene’s test were used to confirm that the data was normally distributed and homogenous. Non-normal distributions were Box–Cox transformed for statistical analysis. For normally distributed variables, the statistical significance of differences between two groups was estimated using the paired Student’s *t*-test or the unpaired Student’s *t*-test; for variables that departed from normality, the Wilcoxon singed-rank test or Mann–Whitney U test was applied, respectively. To compare differences between more than two groups, ANOVA for repeated measurements and the post hoc Bonferroni test or Friedman test following the post hoc analysis with Dunn’s multiple comparison test were used. Due to the relatively small sample sizes and the low statistical power of the estimated inferences in most calculations, the resampling bootstrap technique (10,000 iterations) was used to determine the likelihood of obtaining the revealed differences due to pure chance; in such circumstances, the study refers to the bootstrap-boosted test statistics instead of the classical approach. Statistica v. 13.1 (Dell Inc., Tulsa, OH, USA), StatsDirect v.3.0.182 (Merseyside (Birkenhead), UK), GraphPad Prism v.5 (San Diego, CA, USA), and Resampling Stats Add-in for Excel v.4 (Arlington, VA, USA) were used in statistical calculations. 

## 5. Study Limitations

Our data on platelet reactivity in healthy donors under in vitro conditions are based on a small sample size. Nevertheless, the results appear consistent and are further corroborated by the ex vivo data from COVID-19 patients. Furthermore, it is worth mentioning that the differentiation of nAb anti-spike protein antibody concentration in blood donors was too low: only three donors from the total group were negative for anti-spike protein IgG (below 33.8 BAU/mL), and over 50% of donors had more than 2080 BAU/mL. Although the study used the commercial SARS-CoV-2 TrimericS IgG test for determining antibody titer (neutralizing anti-spike protein antibodies; nAb), this assay only measures values from 4.81 to 2080 BAU/mL; as such, exact values for samples higher than 2080 BAU/mL are not available. Secondly, only one concentration of spike protein was used (2 µg/mL); however, this value was selected based on previous in vitro experiments on interactions between blood platelets and spike protein. Thirdly, despite the large number of tests used to assess platelet function, it was not possible to demonstrate a potential mechanism of blood platelet interactions with spike protein or with spike protein/anti-spike protein nAb complex. We can merely suggest that two main platelet surface receptors, GPIb or GPIIbIIIa, may be involved in this process. Hence, further studies are needed to confirm our findings, determine the extent to which the thrombotic complications in COVID-19 depend on the presence of the spike protein itself in blood, and confirm the role of nAb (IgG anti-spike protein neutralizing antibodies) in hemostatic disturbances.

## 6. Conclusions

Platelet reactivity is modulated not only by the SARS-CoV-2 spike protein but also by anti-spike protein-neutralizing antibodies. These factors affect platelet function in different ways, resulting in reduced aggregation or decreased GPIIbIIIa (fibrinogen receptor) activation. The presence of spike protein also induces the binding of vWF to platelets in ristocetin-treated blood and accelerates thrombus retraction. Additionally, spike protein increases the collagen-stimulated aggregation of isolated platelets. Our findings suggest that blood platelet function may be modulated by spike protein and by the spike protein/-anti-spike protein neutralizing antibodies complex. These interactions include GPIIbIIIa or GPIb, but further studies are needed to explore molecular mechanisms. Future studies on platelet activation/reactivity in COVID-19 patients or in donors vaccinated with anti-SARS-CoV-2 and/or previously infected COVID-19 should be supported by measurements of spike protein and anti-SARS-CoV-2 antibody concentrations in blood.

## Figures and Tables

**Figure 1 ijms-24-05312-f001:**
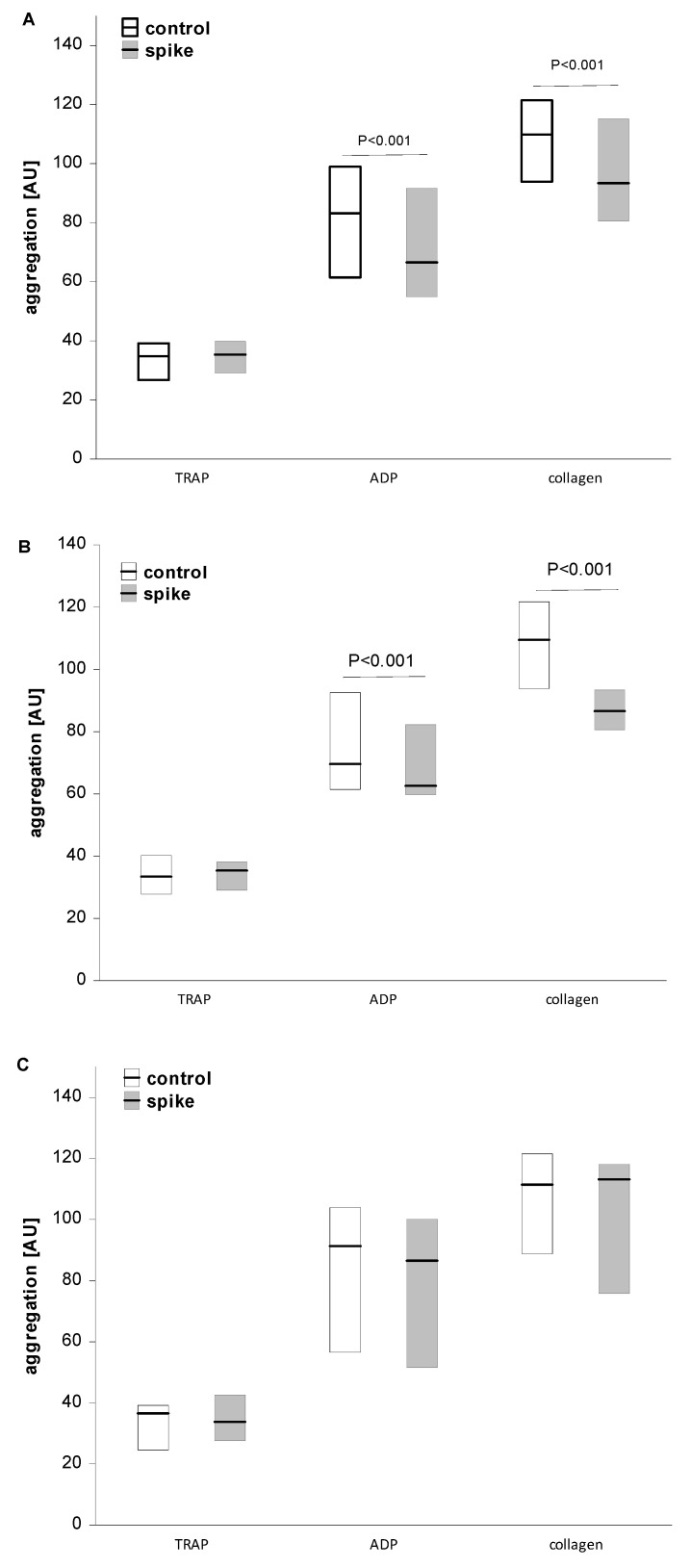
The effect of spike protein on platelet aggregation in whole blood. Data are shown as median and interquartile range (Q1; Q3). The analysis was performed in the total group, n = 38 (**A**), in the High Ab-group, n = 19 (**B**), and in the Low Ab-group, n = 19 (**C**). A significant reduction in the presence of spike protein was observed for ADP- or collagen-induced aggregation in all participants (**A**) and in the High Ab-group (**B**) (*p* < 0.001), but not in the Low Ab-group (**C**). The statistical significance of differences between control vs. spike sample was estimated with the bootstrap-boosted paired Student’s *t*-test.

**Figure 2 ijms-24-05312-f002:**
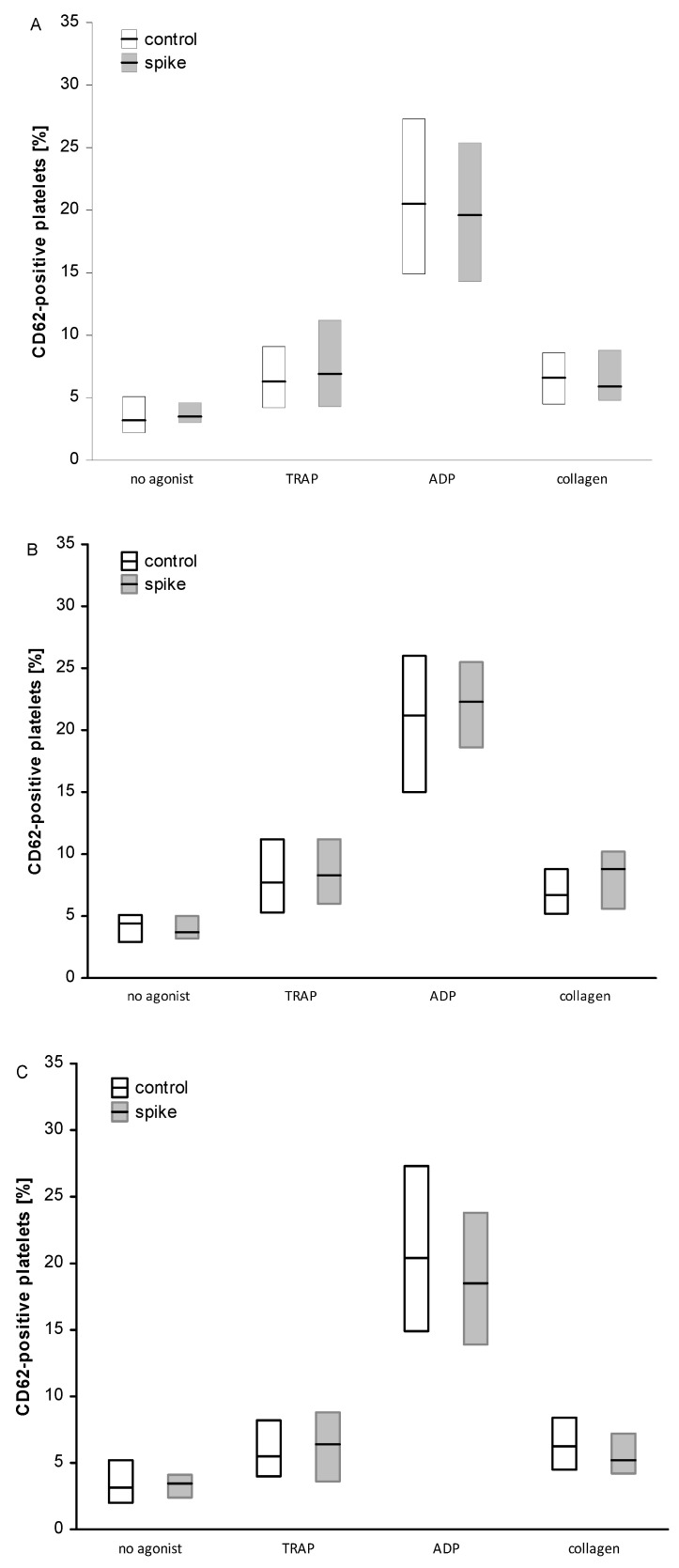
The effect of spike protein on CD62 exposure on the platelet surface membrane. Data are shown as median and interquartile range (Q1; Q3). The analysis was performed in all participants, n = 31 (**A**), in the High Ab-group, n = 13 (**B**), and in the Low Ab-group, n = 18 (**C**). No significant differences were observed between spike and control samples, irrespective of the concentration of nAb or type of agonist. The statistical significance of the differences between control vs. spike sample was estimated with the bootstrap-boosted paired Student’s *t*-test.

**Figure 3 ijms-24-05312-f003:**
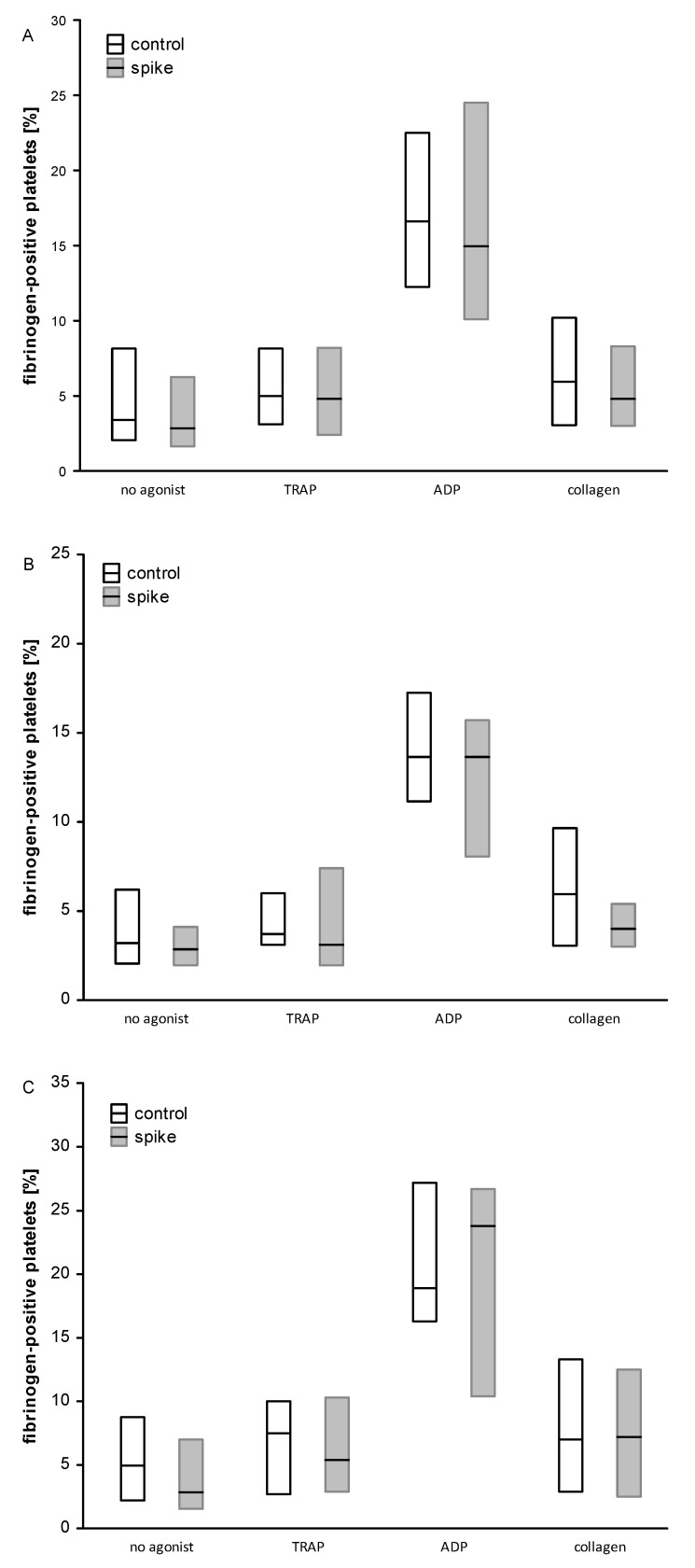
The effect of spike protein on the fibrinogen binding to platelets. Data are shown as median and interquartile range (Q1; Q3). The analysis was performed in all participants, n = 31 (**A**), in the High Ab-group, n = 13 (**B**), and in the Low Ab-group, n = 18 (**C**). No significant differences were observed between spike and control samples, irrespective of the concentration of nAb or the type of an agonist. The statistical significance of the differences between the control and spike samples was estimated with the bootstrap boosted-paired Student’s *t*-test.

**Figure 4 ijms-24-05312-f004:**
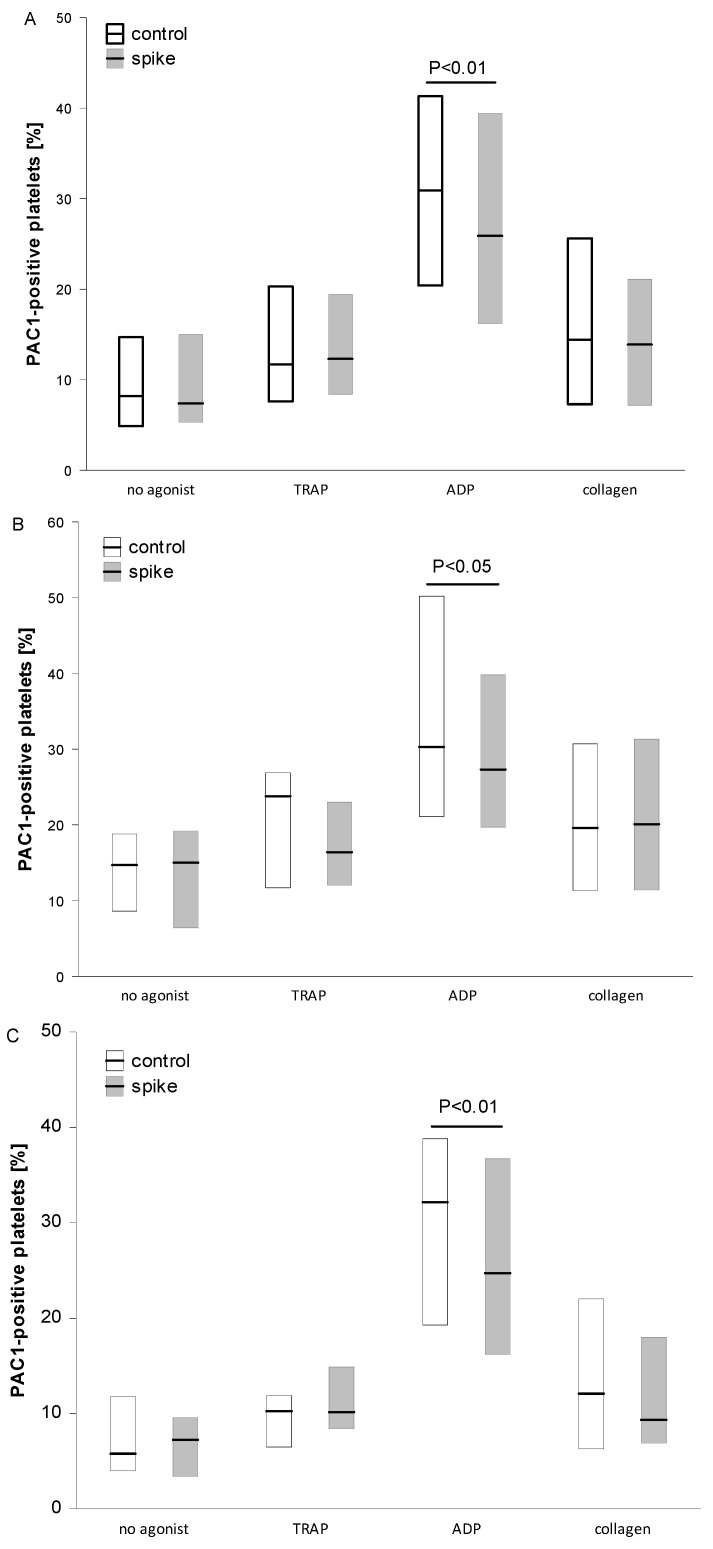
Percent fraction of platelets positive with active fibrinogen receptors (PAC1 binding) in the presence of spike protein. Data are shown as median and interquartile range (Q1; Q3). The analysis was performed in all participant groups, n = 31 (**A**), in the High Ab-group, n = 13 (**B**), and in the Low Ab-group, n = 18 (**C**). A significant reduction in the presence of spike protein was observed for ADP-stimulated platelets irrespective of Ab concentration (*p* < 0.05). The statistical significance of the differences between control vs. spike sample was estimated with the bootstrap-boosted paired Student’s *t*-test.

**Figure 5 ijms-24-05312-f005:**
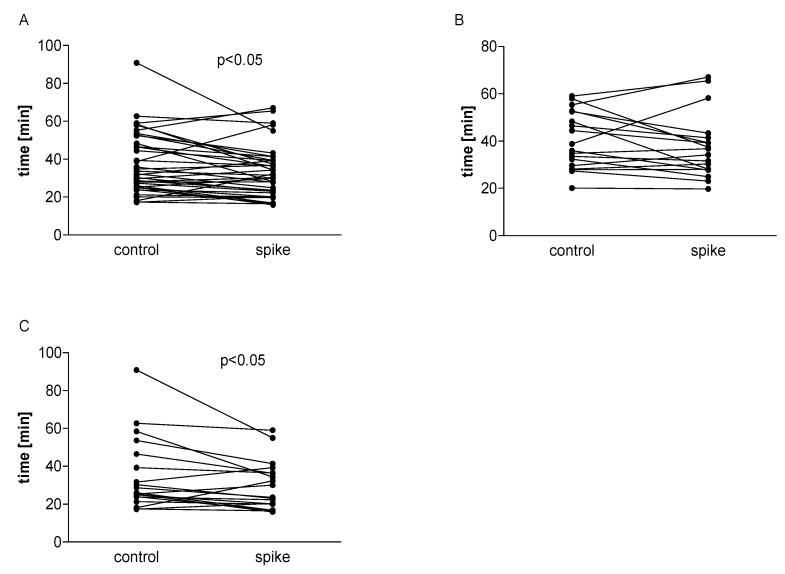
The effect of spike protein on clot retraction. The analysis was performed on the total group of all participant, n = 37 (**A**), in the High Ab-group, n = 18 (**B**), and in the Low Ab-group, n = 19 (**C**). Both the total group (**A**) and the Low- nAb group (**C**) demonstrated significantly shorter times to halving the apparent thrombus area in the presence of spike protein (*p* = 0.017 for total group, *p* = 0.029 for Low- nAb group). The statistical significance of the differences between control vs. spike sample was estimated with the bootstrap-boosted paired Student’s *t*-test.

**Figure 6 ijms-24-05312-f006:**
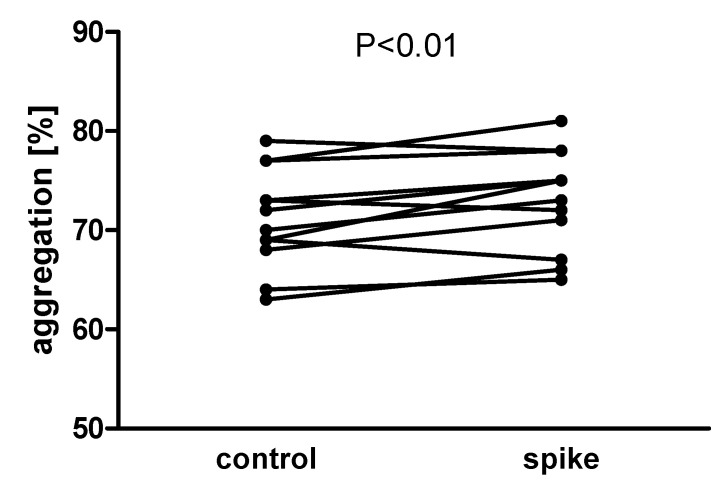
Collagen-induced aggregation of isolated platelets in the presence of the spike protein. The spike protein significantly increased platelet aggregation compared to the control (*p* < 0.01; n = 13). The statistical significance of differences between control vs. spike sample was estimated with the bootstrap-boosted paired Student’s *t*-test.

**Figure 7 ijms-24-05312-f007:**
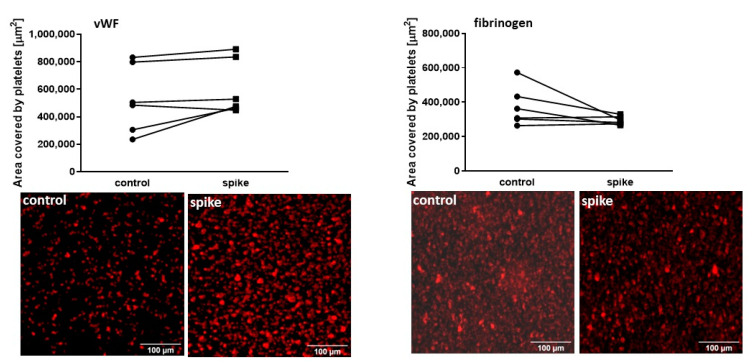
The platelet adhesion under flow conditions to vWF and fibrinogen in the presence of spike protein. No significant differences were observed between spike-treated and control samples (n = 6). Representative images of platelets stained against CD41. Statistical significance of differences between control vs. spike sample was estimated with the Wilcoxon signed-rank test.

**Figure 8 ijms-24-05312-f008:**
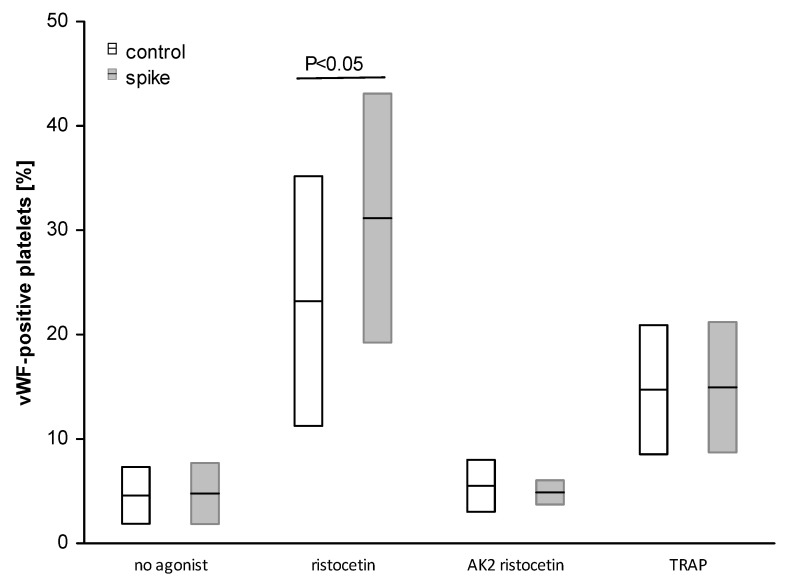
The binding of plasmatic vWF to the platelet surface in the presence of spike protein. Data are shown as mean ± SD. The vWF bound to platelets was identified with anti-vWF/FITC by using flow cytometry (more details in M&M). Spike protein significantly increased vWF binding to platelets compared to control in ristocetin-treated blood, *p* < 0.05 (bootstrap-boosted paired Student’s *t*-test, n = 13). Ristocetin (n = 13) and TRAP (n = 6) significantly increased the vWF-positive fraction of platelets in controls (*p* < 0.05) and in spike-treated samples (*p* < 0.01) compared to non-agonist samples. AK2 antibodies significantly reduced the vWF-positive fraction of platelets compared to ristocetin alone (AK2 ristocetin vs. ristocetin for control, *p* < 0.05, n = 6; AK2 ristocetin vs. ristocetin for spike, *p* < 0.001, n = 6). The significance of the differences between no agonist vs. agonist-treated samples was estimated with the ANOVA for repeated measurements and the post hoc Bonferroni test.

**Table 1 ijms-24-05312-t001:** Characteristics of the study participants enrolled for monitoring of platelet reactivity depending on anti-spike protein nAb concentration.

	All Study Participants n = 38	High Ab-Group n = 19	Low Ab-Group n = 19
Age [years]	29.0 (26.0; 37.0)	31.0 (27.0; 41.0)	27.0 (23.0; 37.0)
BMI [kg/m^2^]	22.3 (20.6; 25.6)	23.1 (20.8; 25.6)	21.7 (19.5; 25.9)
PLT [×10^3^/µL]	257 (218; 301)	260 (206; 308)	255 (220; 284)
MPV [fl]	10.6 (10.1; 11.4)	10.3 (9.9; 11.4)	10.9 (10.1; 11.5)
RBC [×10^6^/µL]	4.7 (4.3; 5.0)	4.7 (4.3; 5.1)	4.7 (4.5; 5.0)
WBC [×10^3^/µL]	5.7 (4.7; 6.9)	5.4 (4.6; 7.3)	5.8 (5.2; 6.7)
NEU [×10^3^/µL]	2.8 (2.4; 3.9)	2.9 (2.4; 3.9)	2.8 (2.4; 3.8)
LYM [×10^3^/µL]	2.1 (1.7; 2.4)	1.9 (1.7; 2.3)	2.3 (1.7; 2.5)
PLR	124 (100; 158)	129 (105; 161)	107 (98; 156)
MPVLR [fl/10^9^/L]	5.0 (4.3; 6.2)	5.1 (4.5; 6.2)	4.7 (4.1; 6.2)
NLR	1.4 (1.1; 1.7)	1.6 (1.2; 1.8)	1.4 (1.0; 1.6)
CRP [mg/mL]	1.0 (0.5; 2.3)	0.9 (0.5; 2.1)	1.1 (0.4; 2.7)
nAb [BAU/mL]	1115 (204; 2080)	2080 (2080; 2080)	204 (55; 831) ***

Data presented as median and quartiles (Q1; Q3). Significance of differences between High Ab-group vs. Low Ab-group was estimated using the Mann–Whitney U test, *** *p* < 0.0001. No statistically significant differences were found. BMI—body mass index; PLT—platelets; RBC—red blood cells; WBC—white blood cells; NEU—neutrophils; LYM—lymphocytes; PLR—platelet-to-lymphocyte ratio; MPVLR—MPV-to-lymphocyte ratio; NLR—neutrophil-to-lymphocyte ratio; CRP—C-reactive protein; nAb–anti-spike protein IgG neutralizing antibodies.

**Table 2 ijms-24-05312-t002:** Fractions of positive platelets with CD62 exposure and PAC1 bound to activated fibrinogen receptor (GPIIbIIIa) after induction with agonists at high concentration in High and Low Ab-group.

	High Ab-Group (n = 13)	Low Ab-Group (n = 18)
PAC1 fraction [%]		
no agonist	6.3 (4.0; 9.9)	5.2 (2.2; 7.7)
Collagen 10 µg/mL	40.3 (30.6; 50.8)	41.5 (21.3; 58.0)
ADP 10 µM	77.7 (63.0; 81.0) *	62.6 (50.2; 72.4)
TRAP 8 µM	57.0 (52.1; 75.1)	51.4 (25.7; 70.8)
CD62 fraction [%]		
no agonist	1.6 (1.5; 2.7)	1.8 (1.1; 2.4)
Collagen 10 µg/mL	16.6 (11.3; 26.2)	20.2 (12.9; 34.9)
ADP 10 µM	52.7 (47.0; 55.0)	41.6 (36.1; 58.9)
TRAP 8 µM	71.9 (59.5; 80.1)	63.4 (45.6; 78.5)

Data presented as median and quartiles (Q1; Q3). Significance of differences between High Ab-group vs. Low Ab-group was estimated using the Mann–Whitney U-test, * *p* < 0.05.

**Table 3 ijms-24-05312-t003:** The influence of spike protein on collagen-induced whole blood platelet aggregation in the presence of anti-GPIb (AK2) antibodies.

	Control	Spike	Control + AK2	Spike + AK2
Aggregation [AU]	91.9 ± 30.2	79.8 ± 20.0 *	89.9 ± 29.9	72.0 ± 27.7 #

Data are shown as mean ± SD, n = 7. The spike protein significantly reduced platelet aggregation in control sample (* *p* < 0.05; spike vs. control) and in blood preincubated with anti-GPIb (AK2) antibodies (# *p* < 0.01; spike + AK2 vs. control + AK2). The statistical significance of differences between non-agonized vs. agonist-treated samples was estimated with ANOVA for repeated measurements and the post hoc Bonferroni test.

## Data Availability

The data presented in this study are available on request from the corresponding author.

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
