# Peer review of "SARS-CoV-2 Spike Protein and Neutralizing Anti-Spike Protein Antibodies Modulate Blood Platelet Function"

_ijms, 2023, doi:10.3390/ijms24065312_

Round 1

Reviewer 1 Report

This article focuses on platelet activation by SARS-CoV-2 Spike protein and the role that anti SARS-CoV-2 neutralizing antibodies play in platelet-Spike protein interactions and platelet activation/aggregation. The article presents novel information and it is well written, results are presented clearly and discussion is relevant and interesting. This article provides new outputs on platelet activation by SARS-CoV-2 considering the presence of neutralizing antibodies due to COVID-19 or vaccines, nevertheless there are important subjects to be addressed: 

I suggest that flow cytometry graphs & plots should be presented in MFI (mean fluorescence intensity) instead of AU. It is difficult to understand graphs in these units.

Why were 2 mg Spike /15 min used as Spike protein concentration and stimulation time? Several articles describe activation peak after 30 min incubation. These protein concentrations and incubation times were selected after performing a kinetics curve? Please state how experiment conditions were selected.

How many experiments were performed from each condition?

Please state PAC1 activity instead of expression.  The receptor is expressed, what the antibody recognizes is the activation of the receptor.

Could His Tag label induce platelet activation by itself? Could the activation observed be a result of the label and not the spike protein itself? Please provide information about this subject.

This article reports increased platelet activation in the positive antibody group, this is not found in the group with no antibodies. So the activation could be due to something in the microenvironment, or the ag-ab complexes? This is a sensitive issue as all vaccinated subjects also have anti-spike antibodies.

Minor Issues

Line 401 and others in the text: P<0.001, not necessary to state “or less”, this is already indicated by the “<” sign.

Some font and font size changes are found through the manuscript: (v.gr, line 188-189, 249-250, 286-290) please check all manuscript to be consistent.

Author Response

Thank you for your letter and Reviewer’s comments concerning our manuscript entitled “SARS-CoV-2 spike protein and neutralizing anti-spike protein antibodies modulate blood platelet function”. Those comments were all valuable and very helpful for the revising and improving of our paper, as well as were the important guiding significant for researchers. We have studied the comments carefully and have made corrections, which - we hope, would acquire approval. The revised portions are highlighted in red in the reviewed manuscript. In addition, the point-by-point responses to the comments are listed below.

This article focuses on platelet activation by SARS-CoV-2 Spike protein and the role that anti SARS-CoV-2 neutralizing antibodies play in platelet-Spike protein interactions and platelet activation/aggregation. The article presents novel information and it is well written, results are presented clearly and discussion is relevant and interesting. This article provides new outputs on platelet activation by SARS-CoV-2 considering the presence of neutralizing antibodies due to COVID-19 or vaccines, nevertheless there are important subjects to be addressed: 

I suggest that flow cytometry graphs & plots should be presented in MFI (mean fluorescence intensity) instead of AU. It is difficult to understand graphs in these units.

RE: According to Reviewer’s suggestion the fluorescence intensity values for flow cytometry measurements were presented in tables and figures in Supplementary materials. However, we did not drop the idea of the flow cytometry data presentation also as percent fractions, simply because this measure appeared much more discriminative for our data than median MFI or geoMFI. Also, see the comments to the question #3 by the Reviewer 2. We chose the median of fluorescence intensity as the representative data because our examined antigens' fluorescence histograms clearly departed from normal distributions.

Why were 2 mg Spike /15 min used as Spike protein concentration and stimulation time? Several articles describe activation peak after 30 min incubation. These protein concentrations and incubation times were selected after performing a kinetics curve? Please state how experiment conditions were selected.

RE: In our study, the experimental conditions were selected based mainly on the article of Zhang et al. (Zhang S, Liu Y, Wang X, Yang L, Li H, Wang Y, et al. SARS-CoV-2 binds platelet ACE2 to enhance thrombosis in COVID-19. J Hematol Oncol. 2020;13(1):120). They reported that spike protein enhanced agonist-induced platelet reactivity in a dose-dependent manner. The maximal activation effect of spike protein was similar at its 1 or 2 µg/ml after 5 min preincubation. We used 2 µg/ml and extended the incubation time to 15 min.

How many experiments were performed from each condition?

RE: The exact number of repeats (number of participants) for every experiment is written in tables or in figure legends in the amended version of the manuscript. To make sure that despite of rather low sample sizes we minimized the risk of too hastily rejecting of null hypothesis, we used as a standard the bootstrap (resampling) algorithms instead of simple classical calculus of test statistics.

Please state PAC1 activity instead of expression.  The receptor is expressed, what the antibody recognizes is the activation of the receptor.

RE: Thank you for your comment. The statement was changed in the new version of the manuscript, and instead of using the wording “PAC1 expression”, we used “Percent fraction of platelets with activated fibrinogen receptors (based on the bonding of PAC1 MoAbs recognizing the activated form of GPIIbIIIa)”.

Could His Tag label induce platelet activation by itself? Could the activation observed be a result of the label and not the spike protein itself? Please provide information about this subject.

RE: Thank you for your accurate comment. According to the publication of Lin et al. (“Tag-Free SARS-CoV-2 Receptor Binding Domain (RBD), but Not C-Terminal Tagged SARS-CoV-2 RBD, Induces a Rapid and Potent Neutralizing Antibody Response”, Vaccines (Basel), 2022 Oct 30;10(11):1839), the addition of a polyhistidine tag on a recombinant protein may significantly impair protein function and immunogenicity. This issue is very important but is not commonly considered to have an influence on the structure and function of a (recombinant) protein. Additionally, there is no evidence in the scientific literature that polyhistidine sequence (HisTag) influences human blood platelet function. In our experiments, we did not use spike protein not labeled with HisTag as a control. We assumed that the HisTag recombinant spike protein possesses similar activity towards blood platelets as a native spike protein, because the presence of tags of such a small size in large molecule of trimeric spike protein seems rather unlikely to alter protein functionality. However, in our future study on spike protein and platelets, we will certainly use the HisTag-free protein as a control for the HisTag protein.

This article reports increased platelet activation in the positive antibody group, this is not found in the group with no antibodies. So the activation could be due to something in the microenvironment, or the ag-ab complexes? This is a sensitive issue as all vaccinated subjects also have anti-spike antibodies.

RE: The evidence of our study suggests that the high level (over 1115 BAU/ml) of anti-spike protein neutralizing antibodies (nAb) may increase platelet reactivity induced with the agonists such as ADP, TRAP or collagen. This effect was observed only for the activation of fibrinogen receptor GPIIbIIIa, but not for CD62 expression, fibrinogen binding, and aggregation, which may suggest that nAb influences the early phase of platelet activation. Our study is rather preliminary and should be continued to explain this issue. As it was described in the Limitations of the study in the manuscript, in our study the differentiation of nAb anti-spike protein antibody concentration in blood donors was too poor: only three donors from the total group were negative for anti-spike protein IgG (below 33.8 BAU/ml), and over 50% of donors had more than 2080 BAU/ml. In addition, we conclude based on our results, that the complex of nAb-spike protein may rather decrease platelet aggregation or the fibrinogen receptor activation.

Minor Issues

Line 401 and others in the text: P<0.001, not necessary to state “or less”, this is already indicated by the “<” sign.

RE: The description was corrected in the amended version of the manuscript.

Some font and font size changes are found through the manuscript: (v.gr, line 188-189, 249-250, 286-290) please check all manuscript to be consistent.

RE: The formatting of the text was improved in the new version of the manuscript.

Reviewer 2 Report

In the manuscript “SARS-CoV-2 spike protein and neutralizing anti-spike protein 2 antibodies modulate blood platelet activity” Luzak et al. study the effect of SARS-CoV2 spike protein and of neutralizing anti-spike protein antibodies on platelet reactivity. While this is of potential interest, the paper fails to add convincing new insight and does not help further in understanding how sars-cov2 spike protein modulates platelet reactivity. The data presented is rather confusing, the conclusions seem overstated.

The introduction reads lengthy. It is more a list of literature than a good introduction to the manuscript. A focus on the essential literature essential to the manuscript would be desirable. The results section especially the order of single experiments is quite difficult to follow. The rational for most of the experiments is either missing or not well introduced. The manuscripts starts analyzing differences between a antibody high and low group, however – without stating any reasons- this is not followed up in the fig. 6-8, table3 and the Ab levels in the analyzed group here are not stated.

·        Total group: Please specify: Is this the pooled data of group A (high Ab group) and B (low Ab group)? If so, then this group does not give any further insight compared to A or B alone. Please interpret and discuss the data accordingly.

·        Please provide a gating strategy for the flow cytometric measurements.

·        Flow cytometric data: Table 2: Data is presented as % positive platelets. This does not allow any conclusion on “how” positive the positive population is. Please provide also the (Geo)MFI values as well. This applies all figures dealing with flow cytometric data (Fig. 3-5).

·        How do you explain the differences between PAC-1 + platelets and Fibrinogen-binding to platelets? (especially Fig. 2 ADP vs. Fig. 4 ADP).

·        Figure 1-4, 7: Please specify the box plots (IQR, Mean, Median?). Please provide representative aggregation curves in addition to display lag phase and platelet shape change.

·        The conclusion on nAb influencing platelet reactivity (l 455) is overstated based on the provided results especially regarding the flow data (limited to ADP stimulated, no difference upon stimulation with TRAP and collagen).

·        Fig 5: Please be more precise: p-value was estimated, p<0.05. Please state an exact p-value.

·        Fig. 6: The wording in ll. 487 ff. is misleading. Since there is overall no statistical difference, please rephrase: the platelet adhesion was higher (vWF) / lower (fibrinogen) instead just use “overall unaltered”. Please adapt the rational in l. 502 as well as the interpretation of the data in the discussion section (l. 604) accordingly. The conclusion is overstated in the discussion, since there was overall no difference.

For flow adhesion studies, a quite low n-number was tested. Nothing is stated whether the tested patients belonged to the high/low Ab group. Was there any difference between these groups observed?

The quality of fluorescence images could be improved. The brightness is rather low. A scale bar is missing.

·        Please be more precise: Was the binding of plasmatic vWF to platelets analyzed (as stated in l. 504)? Then: vWF expression on the platelet surface is not a correct wording, rather use: Binding of plasmatic vWF to platelet surface.

·        Fig. 7: Please revise the figure: * and # are not aligned: what do they refer to? Which groups are here compared. This information is missing in the figure legend. The legend of the Bar Graphs is poorly presented. The figure legend is too short: how were vWF positive platelets determined? From which n-number? Which statistical test was used?

·        What was the rational for assessing collagen-induced platelet aggregation in the presence of an anti-GPIba antibody?

·        Fig. 8: Collagen induced aggregation was determined in washed, isolated platelets with the spike protein leading to a slight increase in platelet aggregation. This contradicts the results obtained for whole blood aggregation (Fig. 1), where spike protein led to a decreased platelet aggregation in the total and high Ab group. Similar contradicting results were obtained for ADP (Fig. 8: no effect on aggregation vs. Fig.1 reduced platelet aggregation in the presence of spike protein). These inconsistencies have not been investigated and resolved by further experiment and are not discussed in the discussion section.

Author Response

Thank you for your letter and Reviewer’s comments concerning our manuscript entitled “SARS-CoV-2 spike protein and neutralizing anti-spike protein antibodies modulate blood platelet function”.  Those comments were all valuable and very helpful for the revising and improving of our paper, as well as were the important guiding significant for researchers. We have studied the comments carefully and have made corrections, which - we hope, would acquire approval. The revised portions are highlighted in red in the reviewed manuscript. In addition, the point-by-point responses to the comments are listed below.

In the manuscript “SARS-CoV-2 spike protein and neutralizing anti-spike protein 2 antibodies modulate blood platelet activity” Luzak et al. study the effect of SARS-CoV2 spike protein and of neutralizing anti-spike protein antibodies on platelet reactivity. While this is of potential interest, the paper fails to add convincing new insight and does not help further in understanding how sars-cov2 spike protein modulates platelet reactivity. The data presented is rather confusing, the conclusions seem overstated.

The introduction reads lengthy. It is more a list of literature than a good introduction to the manuscript. A focus on the essential literature essential to the manuscript would be desirable.

RE: According to Reviewer’s suggestion the Introduction has been shortened and modified in the amended version of the manuscript.

The results section especially the order of single experiments is quite difficult to follow. The rational for most of the experiments is either missing or not well introduced. The manuscripts starts analyzing differences between a antibody high and low group, however – without stating any reasons- this is not followed up in the fig. 6-8, table3 and the Ab levels in the analyzed group here are not stated.

RE: Our major aim was to investigate the effects of anti-spike protein neutralizing antibodies (nAb) on the impact of spike protein on platelet function, however, we have made an attempt to elucidate the mechanisms of the above (e.g. a role of GPIb and vWF binding in the spike protein-platelet interactions). On the other hand, the experiments with the isolated platelets were intended to assess the significance of direct interactions between spike proteins and platelets without the contribution of any other blood components, including nAb.

We agree with the Reviewer and realize that the abundance of results and the number of various experimental approaches could pose a problem for the reader. Therefore, we have added some explanatory information and changed the paragraph order in the Results section, which should help to follow the idea of the study.

  • Total group: Please specify: Is this the pooled data of group A (high Ab group) and B (low Ab group)? If so, then this group does not give any further insight compared to A or B alone. Please interpret and discuss the data accordingly.

      RE: Indeed, the total group involves two subgroups: the high nAb-group and low nAb-group. As we have stated in the aim of our study (lines 127-129), “We hypothesize that a high level of anti-spike protein antibodies (nAb) in the blood of healthy donors endures the activating effect of spike protein on platelet reactivity.” Therefore, in order to investigate the role of anti-spike protein neutralizing antibodies on platelet function, the total group was divided into two subgroups based on the median of Ab concentration (median was 1115 BAU/ml). We have added an explanation of the rationale of this division in the Results section in the new version of the manuscript. Obviously, the most important for us was the comparison of platelet function between both the subgroups and the results for the total group were presented in the figures merely for reference. We did not see the reason for statistical comparing of a given subgroup with the whole source group (even ignoring the data pseudoreplication problem).

  • Please provide a gating strategy for the flow cytometric measurements.

RE: The gating strategy for the flow cytometric measurements is described in Materials&Methods in paragraph 2.4. Generally, platelets were gated from whole blood using anti-CD61 antibodies (anti-CD61/PerCp or anti-CD61/PE). In the population of platelets – CD61 positive events, marker-positive platelets (above isotype cut-off) were measured. The analysis of antigen-positive platelet fraction was done in the total CD-61 positive population without discrimination of normal platelets or platelet aggregates in FSC/SSC dot plots. In our opinion, this strategy allows us to simply compare the results from whole blood aggregation and from flow cytometry.

  • Flow cytometric data: Table 2: Data is presented as % positive platelets. This does not allow any conclusion on “how” positive the positive population is. Please provide also the (Geo)MFI values as well. This applies all figures dealing with flow cytometric data (Fig. 3-5).

      RE: In the flow cytometry approach we monitored various types of data including antigen-positive platelet fraction (%) or fluorescence intensity (median MFI as a preferred method to measure MFI of the ‘logarithmically distributed’ histogram). Because we were interested in the estimate of how many platelets (normal platelets or platelets aggregates) were positive for a selected antigen, we have chosen to analyze the percent fractions of antigen-positive platelets (%). Of course, when using the positivity of a cell with regard to any given antigen, we have to be aware that only the objects/cells bearing on their surfaces some minimum number of the copies of a given antigen (in flow cytometry estimated for some 800-1000 copies, depending on the approach) will appraise as antigen-positive cells. Hence, when talking about the antigen-positive cells we understand such a statement in the category of the manifestation of at least 800-1000 antigen copies on cell surface to ensure that a given cell is recorded as the antigen-positive one. Such antigen-positive cells may, of course, bear 1000, 1500, 2000, 10000 or 25000 copies, and to evaluate this fact we need to employ another measure which is fluorescence intensity, relevant to the number of MoAb copies bound to antigen copies on the cell surface.  Thus, using these two measures, we are able to reason on both some threshold manifestation of a given antigen and on its abundance on the surface of a given cell. According to Reviewer’s suggestion, the median values of fluorescence intensity were shown in Supplementary materials. Why we chose the median values instead of geometric mean values is because, in our opinion, this measure better describes fluorescence histograms for the antigens investigated in our study. And why we did not quote them in the earlier version of our MS, is because they simply showed not very high discriminative power in the comparison to the variable of percent fraction positivity.

  • How do you explain the differences between PAC-1 + platelets and Fibrinogen-binding to platelets? (especially Fig. 2 ADP vs. Fig. 4 ADP).

      RE: Thank you for your comment. It seems at the first sight that the data originating from platelet aggregation, fibrinogen binding, and fibrinogen receptor activation (PAC1 exposure) should be fluctuated collinearly, simply speaking – they should correlate with each other. However, the reality opposes this logic. Both the data from this study or from our previous studies, as well as the data reported in scientific literature, do not confirm this statement. The main explanation for this observation is that for the fibrinogen binding measurements, the exogenous fibrinogen labeled with Oregon Green, used always in the same concentration (30 µg/ml), may not necessarily reflect the true endogenous concentrations of fibrinogen in whole blood. Thus, the proportions of plasmatic and added exogenous fibrinogen is different in blood samples and do not adequately mimic/reflect the native/natural conditions.

  • Figure 1-4, 7: Please specify the box plots (IQR, Mean, Median?). Please provide representative aggregation curves in addition to display lag phase and platelet shape change.

      RE: The manner of data presentation is described in the figure and table legends, and the specification of box plots (median with quartiles or mean with standard deviation) is given, too. Also, according to the Reviewer’s suggestion, the representative aggregation curves were included in the Supplementary materials (Figure S4).

     The conclusion on nAb influencing platelet reactivity (l 455) is overstated based on the provided results especially regarding the flow data (limited to ADP stimulated, no difference upon stimulation with TRAP and collagen).

      RE: According to Reviewer’s suggestion the conclusion was changed in the amended version of the manuscript.

  • Fig 5: Please be more precise: p-value was estimated, p<0.05. Please state an exact p-value.

      RE: According to the Reviewer’s suggestion, the exact p-value was included in the legend of Figure 5.

  • Fig. 6: The wording in ll. 487 ff. is misleading. Since there is overall no statistical difference, please rephrase: the platelet adhesion was higher (vWF) / lower (fibrinogen) instead just use “overall unaltered”. Please adapt the rational in l. 502 as well as the interpretation of the data in the discussion section (l. 604) accordingly. The conclusion is overstated in the discussion, since there was overall no difference.

For flow adhesion studies, a quite low n-number was tested. Nothing is stated whether the tested patients belonged to the high/low Ab group. Was there any difference between these groups observed?

The quality of fluorescence images could be improved. The brightness is rather low. A scale bar is missing.

RE: According to the Reviewer’s suggestion the description of data from the adhesion analysis was changed and the appropriate comments were removed from the discussion and abstract. Also, the information about the nAb level in the investigated group was included in the description. Because of the low number of participants in the experiment, we did not subdivide the population studied into two groups according to the nAb level.

The quality of fluorescence images has been improved and a scale bar has been included in an amended version of the figure.

  • Please be more precise: Was the binding of plasmatic vWF to platelets analyzed (as stated in l. 504)? Then: vWF expressionon the platelet surface is not a correct wording, rather use: Binding of plasmatic vWF to platelet surface.

RE: According to the Reviewer’s suggestion, the phrase “vWF expression on the platelet surface” was replaced with “the binding of plasmatic vWF to platelet surface”.

  • Fig. 7: Please revise the figure: * and # are not aligned: what do they refer to? Which groups are here compared. This information is missing in the figure legend. The legend of the Bar Graphs is poorly presented. The figure legend is too short: how were vWF positive platelets determined? From which n-number? Which statistical test was used?

RE: The figure (Fig. 7) has been revised according to the Reviewer’s suggestion. The symbols for statistical significance were corrected and the information about statistical tests, n-number, and the determination of platelet vWF has been included in the figure legend.

  • What was the rational for assessing collagen-induced platelet aggregation in the presence of an anti-GPIba antibody?

RE: Thank you for your comment. The use of anti-GPIba antibodies in platelet aggregation was an attempt to explain the role of GPIba in the interaction of spike protein with platelets resulting in a significant reduction of collagen-stimulated whole blood aggregation (Figure 1 in the manuscript). Also, the effect of spike protein was most extensive for collagen, so we selected this agonist to further study the process with anti-GPIba antibodies in aggregation. Also, our study was prompted by the publication of Li et al. (Li et al. Platelets mediate inflammatory monocyte activation by SARS-CoV-2 Spike protein. The Journal of clinical investigation. 2022;132(4)), who found that SARS-CoV-2 can activate platelets directly and identified GPIbα as the binding receptor for spike protein.

  • Fig. 8: Collagen induced aggregation was determined in washed, isolated platelets with the spike protein leading to a slight increase in platelet aggregation. This contradicts the results obtained for whole blood aggregation (Fig. 1), where spike protein led to a decreased platelet aggregation in the total and high Ab group. Similar contradicting results were obtained for ADP (Fig. 8: no effect on aggregation vs. Fig.1 reduced platelet aggregation in the presence of spike protein). These inconsistencies have not been investigated and resolved by further experiment and are not discussed in the discussion section.

RE: In our study we aimed to determine whether the presence of anti-spike protein antibodies offers protection to blood platelets against the deleterious effects of spike protein. Thus, we estimated the effect of spike protein mainly in whole blood from donors who had (or not) the anti-spike protein neutralizing antibodies (nAb). We concluded that in people with a high level of nAb and with spike protein present in a blood the platelet reactivity can be reduced probably by a generation of nAb-spike protein complex and its interaction with platelets. The experiment with washed platelets was performed in order to check the direct interaction of spike protein with platelets when blood cells or plasma protein were absent. The opposite results from whole blood and washed platelets suggest that spike protein per se may increase platelet reactivity but the presence of nAb against spike protein may reverse this effect.

Round 2

Reviewer 2 Report

The manuscript has improved a lot and the authors addressed my comments.

There is two wording inaccuracies in the paper which need to be fixed:

- always state P-selectin exposure and not expression (this varies within the manuscript and exposure is the correct term)

- In their experiments, the authors did not study PAC-1 expression, but PAC-1 binding (PAC-1 is an antibody binding to the activated form of GPIIb/IIIa). This needs to be corrected throughout the manuscript.

Author Response

Thank you very much for the comments. According to the suggestion the wording "P selectin exposure or CD62 exposure" instead of "P selectin (CD62) expression" and "PAC1 binding" instead 'PAC1 exposure" were included in the amended version of the manuscript. The changes are marked up using the “Track Changes” in the Word document.